# Crystalline TiO$_2$ protective layer with graded oxygen defects for efficient and stable silicon-based photocathode

Jianyun Zheng[1,2], Yanhong Lyu[1,2], Ruilun Wang[1], Chao Xie[1], Huaijuan Zhou [3], San Ping Jiang[2] & Shuangyin Wang[1]

The trade-offs between photoelectrode efficiency and stability significantly hinder the practical application of silicon-based photoelectrochemical devices. Here, we report a facile approach to decouple the trade-offs of silicon-based photocathodes by employing crystalline TiO$_2$ with graded oxygen defects as protection layer. The crystalline protection layer provides high-density structure and enhances stability, and at the same time oxygen defects allow the carrier transport with low resistance as required for high efficiency. The silicon-based photocathode with black TiO$_2$ shows a limiting current density of ~35.3 mA cm$^{-2}$ and durability of over 100 h at 10 mA cm$^{-2}$ in 1.0 M NaOH electrolyte, while none of photoelectrochemical behavior is observed in crystalline TiO$_2$ protection layer. These findings have significant suggestions for further development of silicon-based, III–V compounds and other photoelectrodes and offer the possibility for achieving highly efficient and durable photoelectrochemical devices.

[1] State Key Laboratory of Chem/Bio-Sensing and Chemometrics College of Chemistry and Chemical Engineering, Hunan University, Changsha 410082 Hunan, China. [2] Western Australian School of Mines: Minerals, Energy and Chemical Engineering and Fuels and Energy Technology Institute, Curtin University, Perth, Western Australia 6102, Australia. [3] State Key Laboratory of High Performance Ceramics and Superfine Microstructure, Shanghai Institute of Ceramics, Chinese Academy of Sciences, Shanghai 200050, China. Correspondence and requests for materials should be addressed to H.Z. (email: huaijuanzhou518@mail.sic.ac.cn) or to S.W. (email: shuangyinwang@hnu.edu.cn)

Photoelectrochemical (PEC) water splitting that directly converts water and sunlight to hydrogen and oxygen is an innovative and sustainable approach[1–4]. Much effort has been directed towards searching efficient and stable semiconductor materials for PEC cells, including silicon (Si)[5–10], III–V compounds[11,12], and various oxides[13,14]. In comparison to oxides and III–V semiconductors, Si is more attractive because of its low cost, low bandgap (~1.1 eV), and applicable conduction band edge position for hydrogen evolution reaction (HER)[15–18]. In order to avoid backscattering of light in the planar water–silicon interface and maximize conversion of solar energy to H2 fuel, Oh et al employed Si with nanoporous structure (black Si) comprising the density-graded surface and reported a near-zero reflection[19]. Unfortunately, according to Pourbaix diagrams, black Si is easy to be corroded quickly when contacting with an electrolyte of high ionic strength, meaning a very narrow window of stability[20].

Facing above challenge, the prevailing strategy is to introduce an amorphous insulating coating with high light transmission, extremely high uniformity, and moderate charge conductivity to separate the Si surface from the electrolyte[21]. This has been met with some success with a variety of amorphous oxides, like $TiO_2$[22] and $NiO_x$[23]. The early studies showed the stable Si photocathodes for HER in acidic electrolytes by depositing amorphous oxides. Nevertheless, at the device level, the photocathode and photoanode are commonly placed in the alkaline electrolytes, where the earth-abundant, efficient oxygen evolution reaction (OER) catalysts operate sustainably[24].

However, there is an obvious trade-off between photoelectrode stability and efficiency when amorphous oxide serves as a protective layer. Thin amorphous materials (< 4 nm) possess the passage of high current densities (> 1 A cm$^{-2}$)[7] for efficient photoelectrodes, but, are not stable in this case. To the best of my knowledge, amorphous oxides for stabilizing the Si-based photocathodes in alkaline solutions is less than 100 h[25]. Increasing the thickness of the protective layer cannot solve the problem, as the carrier tunneling probabilities decay exponentially with the layer thickness. In fact, even if the layer thickness can be increased, the poor crystallinity and low-density structure of the amorphous oxides will lead to considerably high solubility and limit the electrochemical stability under strong alkaline conditions during a sustained photoelectrolysis[26]. In contrast to amorphous phase, crystalline transition metal oxides can offer a

higher alkaline tolerance to enhance the long-term stability of Si photocathode; however, their application is hindered by low conductivity. It is known that carrier transport in the amorphous $TiO_2$ can be tuned through changing the concentration and formation of the defects[27,28]. However, there are few reports on tailoring the crystalline oxides to meet the challenges of the protective layer of Si photocathode[29].

In this work, we propose a facile approach to significantly enhance and stabilize the PEC performance of black Si (denoted as b-Si) photocathodes in a strong alkali media (1.0 M NaOH). In this approach, crystalline $TiO_2$ with graded oxygen defects (denoted as black $TiO_2$ or b-$TiO_2$) is used as the protective layer. The graded oxygen defects in the b-$TiO_2$ layer allow photogenerated electrons to flow easily to the surface. Meanwhile, the b-$TiO_2$ layer with crystalline structure is stable and enhances substantially the operational life of b-Si photocathode. Finally, cocatalyst/protective layer/semiconductor architecture presents a saturation current density of ~35.3 mA cm$^{-2}$ and durability of > 100 h at 10 mA cm$^{-2}$ in 1.0 M NaOH.

## Results

**Synthesis of black Si-based photocathodes**. The assembly of b-Si-based photocathodes contained three main steps: chemical etching of Si; crystalline $TiO_2$ protective layer (denoted as c-$TiO_2$); and Pd metal nanoparticles (denoted as Pd NPs, Fig. 1, see Methods and Supplementary Table 1). We investigated the bare b-Si electrode and the one protected by c-$TiO_2$ layers without (c-$TiO_2$/b-Si) and with graded oxygen defects (b-$TiO_2$/b-Si) towards water splitting, respectively. In addition, the concentrations of oxygen defects were varied by changing the deposition time of Ti layer (denoted as b1-$TiO_2$/b-Si and b2-$TiO_2$/b-Si, its preparation conditions listed in Supplementary Table 1). Pd NPs were deposited on the surface as cocatalysts and annealed in vacuum[30]. The Si-based photocathodes with Pd NPs were denoted as Pd/b-Si, Pd/c-$TiO_2$/b-Si, Pd/b1-$TiO_2$/b-Si, and Pd/b2-$TiO_2$/b-Si, respectively.

**Characterization of protective layer/b-Si photocathodes**. X-ray diffraction (XRD) patterns (Fig. 2a) revealed the presence of anatase $TiO_2$ phase in c-$TiO_2$/b-Si, b1-$TiO_2$/b-Si, and b2-$TiO_2$/b-Si. Further analysis of XRD data indicates the presence of an oxygen-deficient $TiO_x$ phase in b1-$TiO_2$/b-Si and b2-$TiO_2$/b-Si,

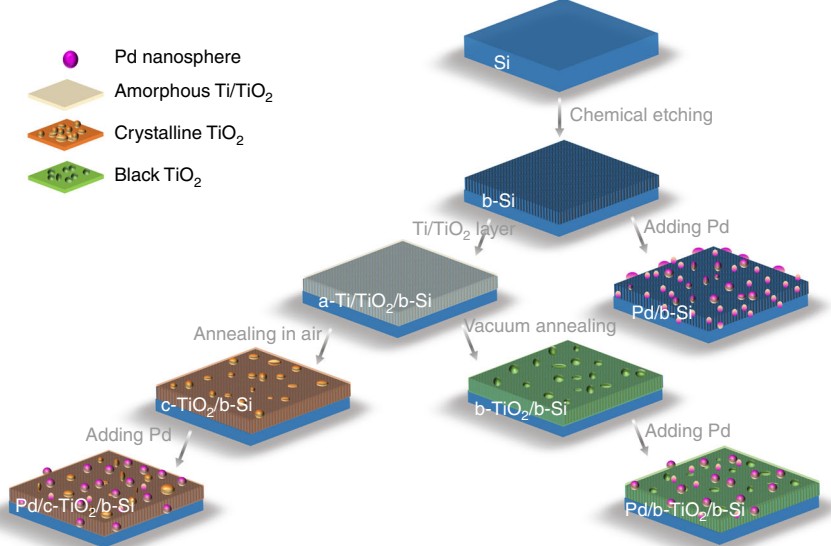

**Fig. 1** Surface protecting strategies for b-Si photocathodes

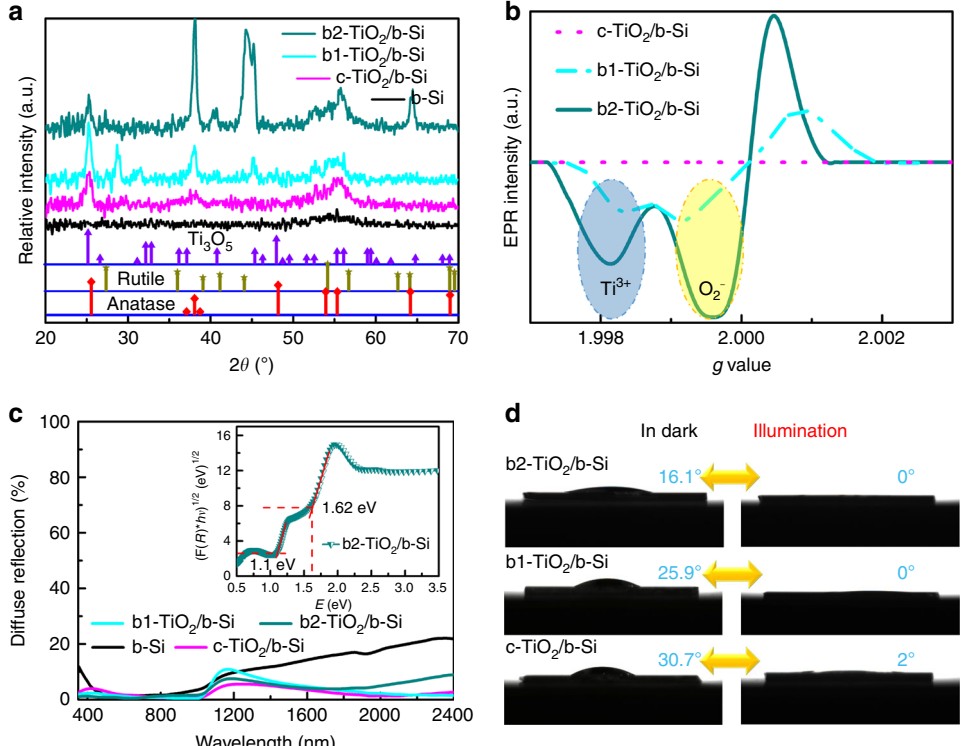

**Fig. 2** Structural, optical and wettability characteristics of the samples. **a** XRD patterns of b-Si (black), c-TiO$_2$/b-Si (pink), b1-TiO$_2$/b-Si (Cambridge blue) and b2-TiO$_2$/b-Si (dark green). The standard XRD patterns for anatase TiO$_2$ (red rhombus), rutile TiO$_2$ (tan star) and Ti$_3$O$_5$ (purple triangle) are shown at the bottom (JCPDS, No. 02–0387, 03–1122 and 23–0606), separately. **b** EPR spectra of c-TiO$_2$/b-Si (pink dot), b1-TiO$_2$/b-Si (Cambridge blue dash) and b2-TiO$_2$/b-Si (green line) at room temperature. **c** The measured total hemispherical optical reflectance of all samples in air. The inset shows the optical absorption coefficient as a function of the incident photon energy for indirect allowed transition for b2-TiO$_2$/b-Si. **d** Photographs of the spherical water droplets with changeable CAs before and after c-TiO$_2$/b-Si, b1-TiO$_2$/b-Si and b2-TiO$_2$/b-Si were exposed to UV-visible light irradiation for 30 min, respectively

because parts of diffraction peaks are in close proximity to Ti$_3$O$_5$ phase. The higher intense peaks at ~44° for b2-TiO$_2$/b-Si can be ascribed to the more b-TiO$_2$ via longer deposition time of Ti layer (Supplementary Table 1). This is supported by an unanticipated peak at ~305 cm$^{-1}$ observed on confocal micro-Raman spectra for both b1-TiO$_2$/b-Si and b2-TiO$_2$/b-Si (Supplementary Fig. 1), which can stem from oxygen deficiencies[15]. The defect feature was further investigated by electron paramagnetic resonance (EPR) spectroscopy at room temperature, as shown in Fig. 2b. Both b1-TiO$_2$/b-Si and b2-TiO$_2$/b-Si exhibited strong EPR signals, but, no signal was detected for c-TiO$_2$/b-Si. The major representations in the spectra ($g$ = ~1.998 and ~2.000) are assigned to paramagnetic Ti$^{3+}$ centers (namely oxygen defects) and reduced O$_2^-$, respectively[31]. The reduced O$_2^-$ generally plays a recombination center of electron-hole pair leading to a low PEC activity. Moreover, b2-TiO$_2$/b-Si had stronger EPR signal than b1-TiO$_2$/b-Si, suggesting higher defect concentration in b2-TiO$_2$/b-Si, in agreement with XRD and Raman data. In such conditions, the oxygen defects are definitely corroborated in b1-TiO$_2$/b-Si and b2-TiO$_2$/b-Si. Field emission scanning electron microscopy (FESEM, Supplementary Fig. 2) indicates that both b-Si and b2-TiO$_2$/b-Si are characterized by a nanoporous structure with non-uniformly distributed pores. The root-mean-square (RMS) roughness of all samples (Supplementary Fig. 3) shows identical surface morphology for the samples.

Figure 2c shows the light diffuse reflection of the samples. Owing to the nanostructured surface trapping light, b-Si displayed distinctly low reflectance (< 5%) in the wavelength range from ~450 to 800 nm. However, the diffuse reflection spectra of b1-TiO$_2$/b-Si and b2-TiO$_2$/b-Si showed a lower reflectance in the 350–1000 nm regions, compared to the bare b-Si and c-TiO$_2$/b-Si. This indicates that the b-TiO$_2$ layer can further increase the light absorption of the samples. Supposing that the light completely absorbed by the samples produces current (unity photoresponse and an infinite thickness of Si), b2-TiO$_2$/b-Si can provide ~37.5 mA cm$^{-2}$ (Supplementary Fig. 4) calculated under the AM1.5 solar photon flux between 350 and 1000 nm. In the inset of Fig. 2c, b2-TiO$_2$/b-Si possesses the two linear parts in the curve, corresponding to two band gaps of ~1.1 and 1.62 eV. The bandgap of ~1.1 eV originating from the Si was also observed in other samples (Supplementary Fig. 5). Furthermore, the optical properties of b-TiO$_2$ film deposited on BK7 glass wafer through the same deposition conditions for b2-TiO$_2$/b-Si illustrate that the b-TiO$_2$ layer shows two band gaps of 1.62 and 2.4 eV (Supplementary Fig. 6). Based on the preparation process and data analysis, a hypothesis is proposed that the b-TiO$_2$ layer in b2-TiO$_2$/b-Si is composed of two kinds of stoichiometric Ti:O (marked as TiO$_{1+y}$ and TiO$_{2-x}$, respectively) as well as graded oxygen defects. As photosensitive materials, the photoinduced superhydrophilicity (water contact angle (CA) of 0°) of c-TiO$_2$ is displayed in Fig. 2d. Among these samples, b2-TiO$_2$/b-Si in dark exhibits most hydrophilic properties in dark with CA of 16.1°. The rich oxygen-deficient surfaces build hydrogen bonds with interfacial water molecules, forming hydrophilic hydration structure[32].

The PEC performances and electronic characterizations of the samples without Pd are provided in Supplementary Figs. 7–11. Among the photocathodes studied, best performance was obtained on b2-TiO$_2$/b-Si, showing the onsets shifted ~0.080 V, ~1.8 mA cm$^{-2}$ at the reversible potential (0 V vs RHE), and

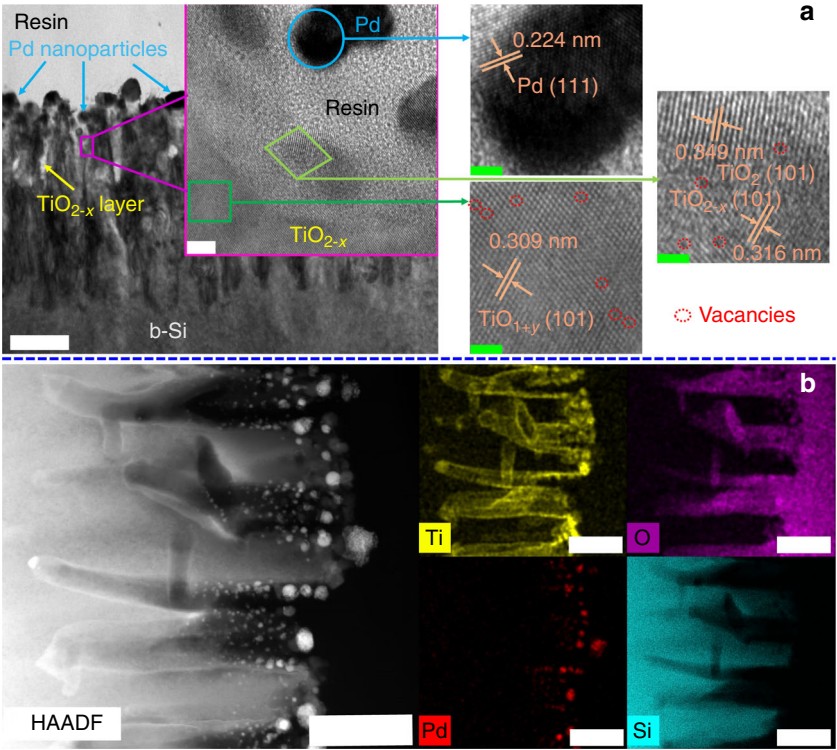

**Fig. 3** Microstructure and composition distribution of Pd/b2-TiO$_2$/b-Si. **a** Cross-sectional TEM image and related high-magnification images of Pd/b2-TiO$_2$/b-Si. The scale bar label is 200 nm in TEM image. The inset in TEM image is the magnification image of the specified area by HRTEM with the scale bar label of 5 nm. High-magnification images on the right correspond to the marked area in the inset with the scale bar label of 2 nm. **b** HADDF-STEM image with matching elemental distribution map for Pd/b2-TiO$_2$/b-Si. Ti, O, Pd, and Si are represented as yellow, purple, red, and Cambridge blue, respectively. All the scale bar labels are 200 nm

limiting current density of 25.2 mA cm$^{-2}$. This is lower than ideal photocurrent of ~37.5 mA cm$^{-2}$ as calculated (Supplementary Fig. 4).

**Characterization and PEC profile of Pd/protective layer/b-Si.** Pd NPs cocatalysts were added to Si-based photocathodes in order to increase the PEC hydrogen production. Pd/b2-TiO$_2$/b-Si showed a sole EPR signal for Ti$^{3+}$ at $g = $ ~1.991, while the one associated with single electron-trapped oxygen vacancies ($g = $ ~2.000) disappeared, which implies that capping Pd NPs restrain the generation of reduced O$_2-$ (Supplementary Fig. 12). Nanoporous structure was maintained in Pd/b1-TiO$_2$/b-Si and Pd/b2-TiO$_2$/b-Si, and some nanoparticles were dispersed on the surface, as seen in Supplementary Figs. 13 and 14. High-resolution transmission electron microscopy (HRTEM) analysis indicates Pd/b2-TiO$_2$/b-Si is made up of Si nanopillars, thin protective layer and nanoparticles (Fig. 3a), in consistent with FESEM images. The spacing of lattice fringes with 0.309, 0.316, 0.349 and 0.224 nm detected from the inner of protective layer to nanoparticles, which can be indexed as the (101) planes of TiO$_{1+y}$, TiO$_{2-x}$ and TiO$_2$, and the (111) plane of face-centered cubic (fcc) structures of Pd, respectively. In addition, there are significant vacant lattice sites in (101) plane of TiO$_{1+y}$ and TiO$_{2-x}$, in accordance with the presence of oxygen defects. The spacing of lattice fringes for the protective layer (b-TiO$_2$) increases gradually from the inner to surface of the layer, implying the formation of graded oxygen defects. The cross-section portion of Pd/b2-TiO$_2$/b-Si (Fig. 3b) was further characterized using scanning transmission electron microscopy (STEM) coupled with energy dispersive x-ray spectroscopy (EDS) mapping. The element mapping indicates that b-TiO$_2$ is conformal via tubular geometry surrounding the surface of needle-shaped b-Si. The minimum

thickness of b-TiO$_2$ layer on the roof of the sample was around 20 nm. Strong O signals were observed from the central region of the tubes, while the edge of the tubes exhibited a weak O signal. The gradient distribution of O element mapping is a result of the presence of graded oxygen defects. As expected, Pd NPs were dispersed on the surface of b-TiO$_2$ layer. The sparse dispersion of Pd NPs will not block the light absorption of b-Si. Meanwhile, the photoinduced superhydrophilicity of the samples was difficult to be observed because of the dispersed Pd NPs (Supplementary Fig. 15).

The composition of each layer of Pd/b2-TiO$_2$/b-Si was analyzed using x-ray photoelectron spectroscopy (XPS) depth profiling measurements via etching (Fig. 4). The full survey scan of the sample surface shows the characteristic peaks of Ti 2p, Pd 3d, and O 1 s (Supplementary Fig. 16). The Pd 3d spectrum of Pd/b2-TiO$_2$/b-Si showed a dominant signature peak of metallic Pd (Pd$^0$). To reveal the variation of the chemical environment of b-TiO$_2$, the Ti 2p spectra of Pd/b2-TiO$_2$/b-Si were analyzed in detail (Fig. 4a, b). The Ti 2p3/2 spectra obtained on the surface and in the bulk (or inner) of the b-TiO$_2$ layer both display two peaks at 459.5 and 457.8 eV, corresponding to Ti$^{4+}$ and Ti$^{3+}$, respectively. However, the intensity of Ti$^{3+}$ in the inner of the b-TiO$_2$ layer is stronger than that detected on the surface of the b-TiO$_2$ layer, whereas the signal of Ti$^{4+}$ becomes weaker in the inner of the b-TiO$_2$ layer. The ratio of Ti$^{3+}$ and Ti$^{4+}$ directly relates to the concentration of oxygen defects and the difference between the surface and inner of the b-TiO$_2$ layer implies the formation of graded oxygen defects. XPS depth profiling of Pd/b2-TiO$_2$/b-Si (Fig. 4c) discloses that the surface is composed of Pd NPs and b-TiO$_2$ layer. The atomic concentration of Pd and O peaked at the sample surface and decreased by increasing the etching depth. The concentration of Pd approached almost zero

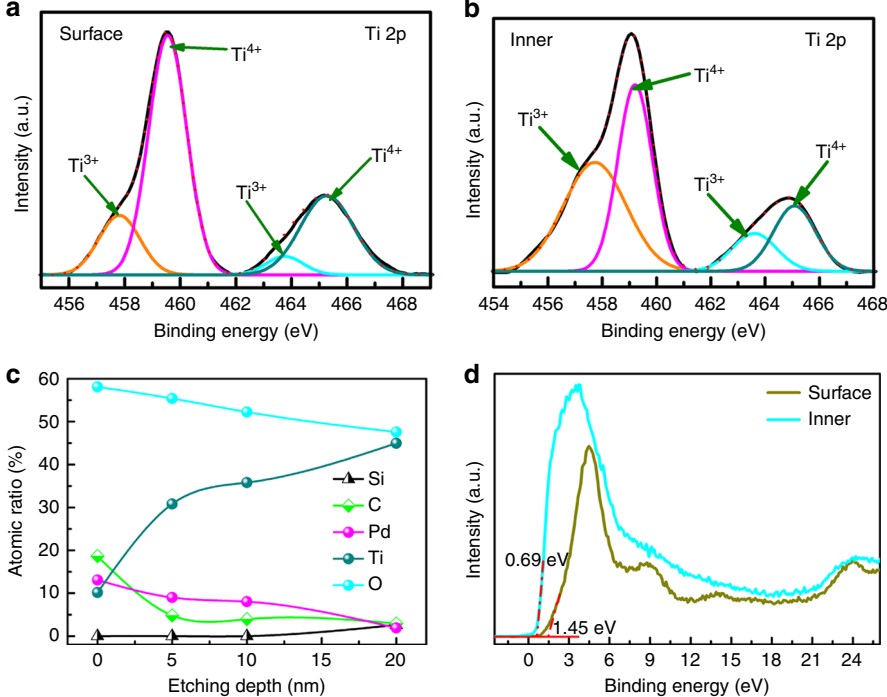

**Fig. 4** XPS depth profiling of Pd/b2-TiO$_2$/b-Si. **a** Ti 2p spectrum for the surface of Pd/b2-TiO$_2$/b-Si. **b** Ti 2p spectrum for the inner of Pd/b2-TiO$_2$/b-Si after the etching depth of 20 nm. **c** Elemental depth profile of Pd/b2-TiO$_2$/b-Si. **d** XPS VB spectra of the surface and inner (etching depth of 20 nm) of Pd/ b2-TiO$_2$/b-Si. It must be mentioned that the extra C 1s peak at 284.8 eV is derived from the carbonaceous contaminant absorbed in the sample surface

with an etching depth of 20 nm. The Ti signal increased continuously and became dominant after the etching depth of 20 nm. The Si signal started to be emerged in XPS spectrum. This indicates that the minimum thickness of b-TiO$_2$ layer for Pd/b2-TiO$_2$/b-Si is around 20 nm, in accordance with STEM images. The results also confirm that b-Si is fully protected in this case. The XPS valence-band (VB) spectra of the surface and the inner of b-TiO$_2$ layer in Pd/b2-TiO$_2$/b-Si (Fig. 4d) demonstrate that the VB maximum of the surface and the inner is 1.45 and 0.69 eV, respectively. This again indicates the formation of the structure of discrete layering for Pd/b2-TiO$_2$/b-Si, defined as Pd NPs/b-TiO$_2$ (or TiO$_{2-x}$/TiO$_{1+y}$)/b-Si concisely.

Figure 5a depicts the PEC performance of HER on Pd/b-Si, Pd/ c-TiO$_2$/b-Si, Pd/b1-TiO$_2$/b-Si, and Pd/b2-TiO$_2$/b-Si under simulated AM 1.5 solar illumination in 1.0 M NaOH solution. The dark currents are negligible. The best performance was obtained on Pd/b2-TiO$_2$/b-Si, achieving an onset potential of 0.32 V vs RHE and photocurrent of ~8.3 mA cm$^{-2}$ at 0 V vs RHE (that is, no thermodynamic bias). This is significantly higher than ~0.17 V and ~4.5 mA cm$^{-2}$ measured on Pd/b1-TiO$_2$/b-Si under identical test conditions. Most interesting, Pd/b2-TiO$_2$/b-Si electrode reached a saturated photocurrent density of ~35.3 mA cm$^{-2}$ at −0.22 V vs RHE, close to the theoretical value of 37.5 mA cm$^{-2}$. This indicates that adding Pd NPs on the sample surface enhances the PEC activity of Si-based photocathodes. On the other hand, Pd/c-TiO$_2$/b-Si showed almost no PEC activity, implying that the Pd NPs do not activate the PEC behaviors of the photocathodes. In addition, Pd/b2-TiO$_2$/b-Si shows a much better PEC performance in 0.5 M H$_2$SO$_4$ electrolyte solution as compared to other samples, similar to that observed in alkaline electrolyte (Supplementary Fig. 17). The high efficiency of Pd/b2-TiO$_2$/b-Si photocathode is supported by much higher incident photon-to-current efficiency (IPCE), as compared to that of Pd/ b1-TiO$_2$/b-Si (Fig. 5b). In the 420−540 nm range, Pd/b2-TiO$_2$/b-Si achieved an IPCE > 90%. Such differences in IPCE between Pd/ b1-TiO$_2$/b-Si and Pd/b2-TiO$_2$/b-Si were also observed in an

acidic solution (Supplementary Fig. 18). Such high IPCE of Pd/ b2-TiO$_2$/b-Si can be explained by (1) good light absorption of b-Si with b-TiO$_2$, and (2) excellent electron transfer owing to high defect concentration. To assess the efficiency of PEC hydrogen generation, long-term hydrogen evolution of Pd/b2-TiO$_2$/b-Si under light illumination was measured at specific potential (−0.078 V) in 1.0 M NaOH. The Faradaic yield (FY) relates to the photocurrent to hydrogen production, and is given by

$$FY = \frac{2n_{H_2}}{It}F \qquad (1)$$

where $F$, $n_{H_2}$, $I$ and $t$ are Faraday's constant, the number of moles of hydrogen gas produced, the photocurrent and the illumination time, respectively. On the basis of the calculation, the FY for hydrogen generation is > 96% on Pd/b2-TiO$_2$/b-Si measured at −0.078 V vs RHE in 1.0 M NaOH under illumination (Supplementary Fig. 19). In addition to IPCE, the external and internal quantum efficiencies (EQE and IQE) are calculated by normalizing IPCE to the measured absorbance of the photocathode and electrolyte,

$$IQE = \frac{EQE}{A} = \frac{EQE}{1-R} \cong \frac{IPCE}{1-R} = APCE \qquad (2)$$

where $A$ is the absorbance of the solution and Pd/b2-TiO$_2$/b-Si. In the wavelength range of 380−750 nm, no absorption of the solution has been verified[33]. Furthermore, IQE is easily represented by absorbed photon-to-current efficiency (APCE) because of the FY of ~1. Here, the APCE value in certain incident photon energy is over 100% (Supplementary Fig. 20). Nevertheless, the solar-to-hydrogen (STH) efficiency and solar-to-hydrogen conversion efficiency (SHCE) of Pd/b2-TiO$_2$/b-Si are low (Supplementary Figs. 21 and 22) because of the lack of p-n junctions.

The durability of Pd/b2-TiO$_2$/b-Si photocathodes were monitored under light irradiation at −0.012 V vs RHE (corresponding

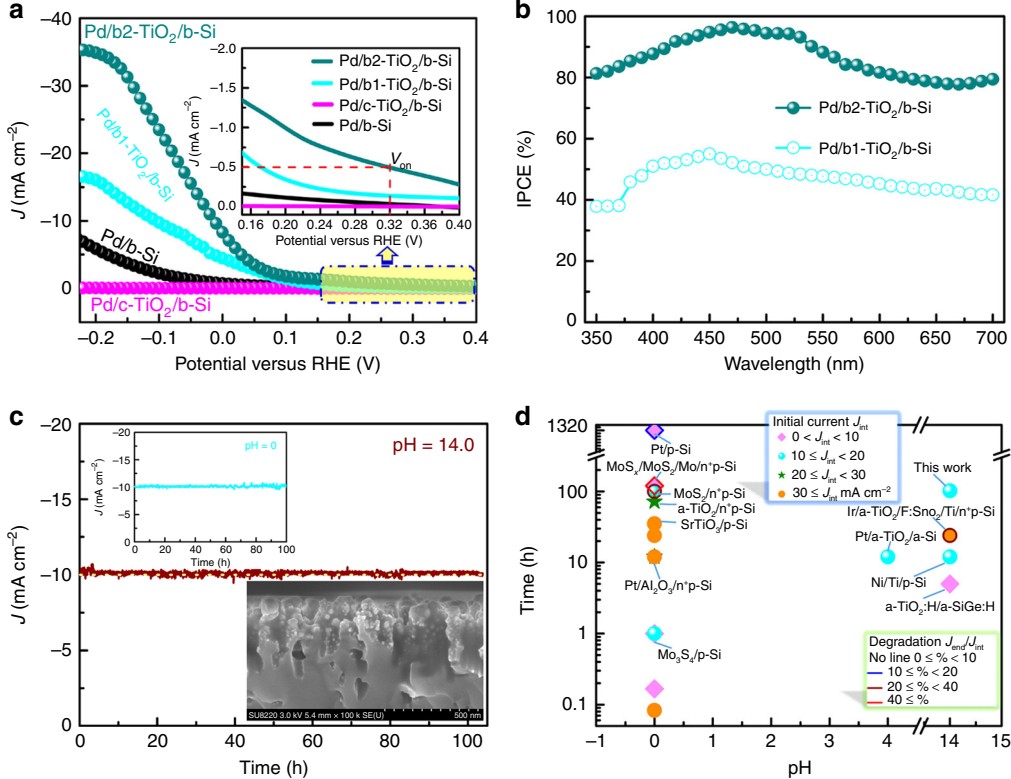

**Fig. 5** PEC performance of Pd/b-Si, Pd/c-TiO$_2$/b-Si, Pd/b1-TiO$_2$/b-Si and Pd/b2-TiO$_2$/b-Si. **a** Current density ($J$)-potential ($V$) curves of all the samples (scan rate is 0.01 V s$^{-1}$) in 1.0 M NaOH under 1 sun illumination. The current density measured in dark is almost horizontal line, namely 0 mA cm$^{-2}$. The inset is the high magnification image corresponding to the marked area. **b** IPCE of Pd/b1-TiO$_2$/b-Si and Pd/b2-TiO$_2$/b-Si in 1.0 M NaOH at −0.22 V vs RHE. **c** $J$-time ($t$) plot of Pd/b2-TiO$_2$/b-Si held at −0.012 V vs RHE in 1.0 M NaOH under 1 sun illumination. The upper-left inset is $J$-$t$ plot of Pd/b2-TiO$_2$/b-Si held at −0.124 V vs RHE in 0.5 M H$_2$SO$_4$ under 1 sun illumination. The bottom-right inset is cross-sectional FESEM image of Pd/b2-TiO$_2$/b-Si after 100 h PEC water splitting in 1.0 M NaOH. **d** Chart visualizing data on stabilities of photocathodes including previous and current work, versus tested pH condition, with resulting photocurrent and degradation rate indicated. Most of device structures for the photocathode with reported stability longer than one hour are noted. $J_{int}$ is the initial photocurrent at the start of the stability test. Degradation rates are calculated using the ratio of the measured photocurrent at the end of the stability test ($J_{end}$) to $J_{int}$

to initial $J$ of ~10 mA cm$^{-2}$) in 1.0 M NaOH and 0.5 M H$_2$SO$_4$ solutions (Fig. 5c). Pd/b2-TiO$_2$/b-Si photocathodes exhibit little photocurrent decay in both alkaline and acid electrolytes for over 100 h (Fig. 5c and Supplementary Fig. 23a). The variations of photocurrent density of Pd/b2-TiO$_2$/b-Si via chopping light illumination at interval time of 2 h (Supplementary Fig. 23b) show good stability and photosensitivity. Meanwhile, similar $J$–$V$ curves of fresh and half-year atmosphere-aged Pd/b2-TiO$_2$/b-Si further verify the stability of the photocathode in air, as shown in Supplementary Fig. 24. Additionally, the b-Si with nanostructured surface remained a high coverage of Pd NPs after 100 h PEC water splitting in 1.0 M NaOH (the inset in Fig. 5c). After a 100-h stability measurement of Pd/b2-TiO$_2$/b-Si, the final Ti and Pd content in the electrolyte (1.0 M NaOH) was evaluated by inductively coupled plasma mass spectrometry (ICP-MS). The results show the dissolution of ~12 ng of Ti and ~19 ng of Pd from the photocathode with a geometric area of ~0.1 cm$^{-2}$. Assuming that the photocathode surface is flat, with a mass loss of ~29 ng Ti per cm$^2$ per day, ~0.3 monolayers of b-TiO$_2$ could be lost every day via operating continuously. Indeed, the actual surface area of Pd/b2-TiO$_2$/b-Si is considerably larger than the geometric area. This implies that the actual b-TiO$_2$ loss is much lower than the calculated value. XPS spectra (Supplementary Fig.25) of Pd/b2-TiO$_2$/b-Si photoelectrode after 100 h photoelectrolysis testify a good level of protection for the photoelectrode surface. However, the atomic concentration of Si at the etching depth of 20 nm after testing is higher than that before testing,

probably due to a small amount of b-TiO$_2$ dissolution in NaOH solution with extended electrolysis. These results highlight the excellent durability of Pd/b2-TiO$_2$/b-Si due to the conformal crystalline TiO$_2$ protective layer. Figure 5d summarizes the stability of current and previous Si-based photocathodes plotted versus the pH level during the test[22,24,27,34–42]. In view of these collected data, most stability studies for the Si-based photocathodes were carried out in the acid electrolytes, specially pH = 0, because most of amorphous protective layers relatively stabilize in such conditions. As mentioned earlier, it is critical to develop the highly stable and efficient photocathodes for HER in strong alkaline electrolytes. In addition, the characterization, analysis and PEC measurements of Pd/b-TiO$_2$/planar Si (substituting planar Si wafer to black Si wafer) further demonstrate the effect of b-TiO$_2$ protection layer (Supplementary Figs. 26–28). In short, the crystalline TiO$_2$ layer with graded oxygen defects supports the p-Si photocathode to achieve a comparable PEC performance and an excellent stability simultaneously.

**Activating and enhancing mechanism by oxygen defects.** In semiconductors, Photoluminescence (PL) emission is a powerful tool to judge the transfer and separation process of the charge carrier. All the samples showed PL peak of approximately 653 nm (Fig. 6a), attributed to the recombination of photogenerated electron-hole pairs for b-Si. Due to coating the c-TiO$_2$ layer, Pd/c-TiO$_2$/b-Si had the strongest PL intensity, in concordance with the poor PEC performance. As compared to either Pd/b-Si or the Pd/

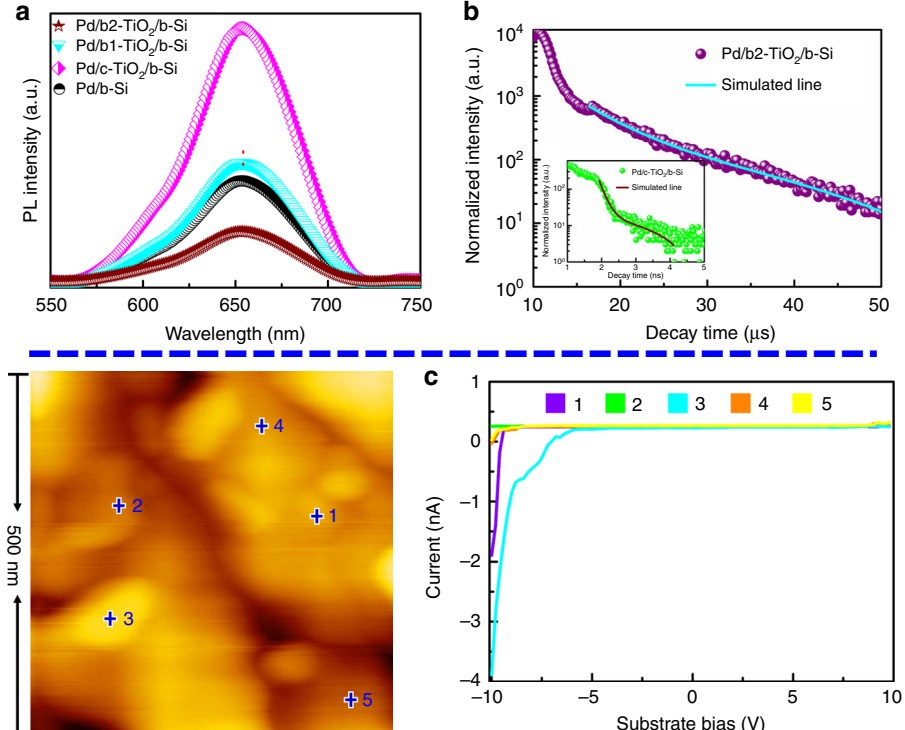

**Fig. 6** Charge carrier dynamics and paths of the samples. **a** Room temperature PL spectra of Pd/b-Si (black), Pd/c-TiO$_2$/b-Si (pink), Pd/b1-TiO$_2$/b-Si (Cambridge blue) and Pd/b2-TiO$_2$/b-Si (brown) with excitation wavelength of 405 nm. **b** Time-resolved PL decay curve for Pd/b2-TiO$_2$/b-Si at room temperature. Solid lines represent the kinetic fit using bi-exponential decay model. The inset is the time-resolved PL decay curve dor Pd/c-TiO$_2$/b-Si. **c** AFM topography image and typical I–V curves of marked position by the number for Pd/b2-TiO$_2$/b-Si. Position 1, 2, 3, 4, and 5 in AFM image corresponding to purple, green, Cambridge blue, orange and yellow I–V curve, respectively

b1-TiO$_2$/b-Si, a pronounced fluorescence quenching was discovered in Pd/b2-TiO$_2$/b-Si, implying that charge carrier recombination is inhibited. Interestingly, the PL intensity decreases with the order of Pd/c-TiO$_2$/b-Si, Pd/b1-TiO$_2$/b-Si, and Pd/b2-TiO$_2$/b-Si, showing the inverse dependence to the concentration of oxygen defects of TiO$_2$ layer. This illustrates the increased charge transfer with oxygen defects in TiO$_2$ layer. Time-resolved PL spectra (Fig. 6b and Supplementary Table 2) were employed to investigate the charge carrier dynamics in Pd/c-TiO$_2$/b-Si and Pd/b2-TiO$_2$/b-Si. Pd/c-TiO$_2$/b-Si displayed quick recombination with PL lifetimes of $t_2 = 8.82$ ns, in accordance with the poor charge transfer capability of c-TiO$_2$. Amazingly, the lifetime of Pd/b2-TiO$_2$/b-Si was prolonged drastically to 16.32 μs, indicating substantially retarded charge recombination and enhanced charge transfer for the Pd NPs/b-TiO$_2$/b-Si photocathode. Moreover, the time-resolved PL spectra of Pd/b-Si and Pd/b1-TiO$_2$/b-Si in Supplementary Fig. 29 and Supplementary Table 2 also presented the level of microsecond lifetime assigning to $t_2 = 13.34$ and 12.54 μs, respectively. Accordingly, more efficient charge transfer and greatly reduced charge recombination are induced by higher concentration of oxygen defects, prolonging the electron lifetime. This indicates that a channel constructed with graded oxygen defects in b-TiO$_2$ layer is facile for transferring the photogenerated electron from the bulk of b-Si to the electrode/electrolyte interface. Conductive atomic force microscopy (C-AFM) allows direct probing of the electron transport channel formed at the different concentrations of oxygen defects. Figure 6c exhibits C-AFM information, including surface topography and typical current–voltage curves for Pd/b2-TiO$_2$/b-Si. Pd/b2-TiO$_2$/b-Si in the positions 1, 3, 4, and 5 showed the current responses. Among them, the position 1 and 3 on the Pd NP exhibited a high current, and the current of the positions 4 and 5

without Pd NP reduced markedly. The current response of Pd/c-TiO$_2$/b-Si (Supplementary Fig. 30) was hardly observed in the range of applied bias, whereas there were only the weak current responses in Pd/b1-TiO$_2$/b-Si (Supplementary Fig. 31). In addition, although the current of Pd/b-Si (Supplementary Fig. 32) was higher than that of Pd/b2-TiO$_2$/b-Si, Pd/b2-TiO$_2$/b-Si could possess more conductive channel than Pd/b-Si. As a result, the graded oxygen defects form the conductive channel would facilitate the electron transfer through the b-TiO$_2$ layer. The concentration of oxygen defects not only determines the passage of the electric current through TiO$_2$ layer, but also influences the distribution and quantity of transport channel.

As mentioned above, the activated and enhanced mechanisms for the PEC performance of Si-based photocathodes with the b-TiO$_2$ layer are proposed in Fig. 7. In general, when illuminated, the b-Si absorbs the incident photons to produce the photoexcited electron-hole pairs. Subsequently, the minority carriers of b-Si tunnel through the protective layer to the Pd NPs, where HER takes place. However, in this work, the crystalline TiO$_2$ layer inhibits the photogenerated electrons to pass through. Graded oxygen defects constructed in the b-TiO$_2$ can form certain electrically conducting channels, which allow the photogenerated electrons to reach on the surface of the photocathode (the left inset of Fig. 7). Additionally, the H$_2$ evolution off the b-TiO$_2$ surface that is similar to hydrogen spillover-assisted H$_2$ evolution off the SiO$_2$ surface[43] is attributed to highly electron conductive and sparsely dispersed Pd NPs. In this reaction process, H$^+$ is reduced for yielding an absorbed proton on the Pd NP, then spillover to the b-TiO$_2$ surface and recombine with another H atom for producing H$_2$. The latter can be accelerated by plentiful electrons tunneling from the b-TiO$_2$ layer immediately to the H$^+$ species on the surface. To further understand the role of b-TiO$_2$

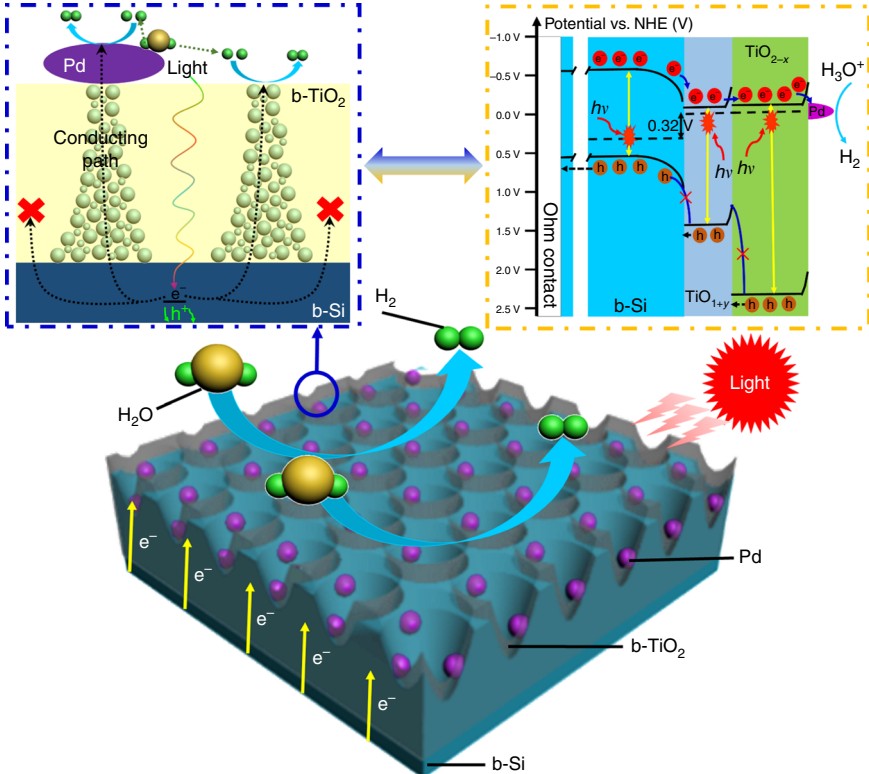

**Fig. 7** Schematic of Pd nanoparticles/black TiO$_2$/b-Si photocathode for water reduction. The left inset (blue rectangular box with dotted line) is the matching side-view of Pd nanoparticles/black TiO$_2$/b-Si photocathode marked by blue ring. The right inset (orange rectangular box with dotted line) is the proposed energy band diagram of Pd nanoparticles/black TiO$_2$/b-Si photocathode

on the transfer of photoexcited electron, the energy band diagram scheme of Pd NPs/b-TiO$_2$/b-Si photocathode is schematically described in the inset of Fig. 7. The VB maximum value in electron volts is converted to electrochemical energy potentials in volts by the reference[44]. In many experimental and computational works, the Ti$^{3+}$ in TiO$_2$ provides a shallow energy level near the conduction band (CB) minimum slightly, while on-lattice disorders in crystalline TiO$_2$ created by oxygen defects can availably up-shift the valence band edge of TiO$_2$[45,46]. In fact, we demonstrate that the tails effect narrows the bandgap of b-TiO$_2$ (1.62 V for TiO$_{1+y}$ and 2.4 V for TiO$_{2-x}$), corresponding to the CB minimum (−0.08 V for TiO$_{1+y}$ and −0.1 V for TiO$_{2-x}$) and VB maximum (1.54 V for TiO$_{1+y}$ and 2.3 V for TiO$_{2-x}$). As a result, the TiO$_{1+y}$ and TiO$_{2-x}$ parts of the b-TiO$_2$ layer have nearly uniform CB edge. In addition, the VB minimum and bandgap of p-type Si generally are at −0.55 and 1.1 V, separately. From the above values, the difference of CB minimum and VB maximum between Si and b-TiO$_2$ are 0.45 and 1.04 V, respectively. Consequently, the electrons go easily through the conduction band, whereas the transport of holes from b-Si to b-TiO$_2$ is blocked due to a large barrier in valance band.

## Discussion

A facile method for resolving the trade-off between efficiency and stability of the Si-based photocathodes is expounded by constructing crystalline TiO$_2$ protective layer with graded oxygen defects. In these cases, graded oxygen defects are the most important factor for activating and enhancing PEC behaviors of Si-based photocathodes with crystalline TiO$_2$ protective layer. Meanwhile, the conformal crystalline TiO$_2$ layer substantially enhances the lifetime of the photocathodes in both strong acid and alkaline electrolyte solutions. The present results illustrate an effective path toward the practical solar-fuel photocatalysts

systems via introducing crystalline nonstoichiometric metal oxide layer.

## Methods

**Preparation of black silicon.** A single-side-polished, boron-doped p-type Si wafer with (100) orientation, a thickness of 500 μm, and a resistivity of 2–4 Ω cm (Supplementary Figs. 33 and 34) was ultrasonically cleaned in the acetone, ethanol, and deionized (DI) water for 20 min, separately. b-Si was fabricated by metal-catalyzed electroless etching method[9,47]. The clean wafer was immersed in a piranha solution (1:3 (v/v), 30 wt% H$_2$O$_2$:H$_2$SO$_4$) for 5 min and then in a 5 wt% HF for 10 min to remove the SiO$_2$ layer. Subsequently, the wafer was rinsed with DI water and immersed in a hybrid solution of 2 mM AgNO$_3$ and 2 wt% HF for 30 s. After that, the wafer was fast rinsed with DI water and subsequently dipped in the solution of 40 wt% HF, 20 wt% H$_2$O$_2$ and DI water with a volume ratio of 3:1:10 for 150 s. Further, rinsing with DI water again, the etched wafer was soaked with 40 wt% HNO$_3$ solution for 20 min to remove the Ag nanoparticle residue. Finally, the prepared wafer with porous surface (b-Si) was rinsed and dried.

**Assembly of black silicon-based photocathodes.** Firstly, amorphous TiO$_2$/Ti multilayer was deposited on b-Si by using a direct current magnetron sputtering (dcMS) system (Chuangshiweina Co. Ltd., MSP-3200) to sputter Ti target (purity > 99.5 wt%) in pure Ar (99.99%) or mixed Ar/O$_2$ (99.99%) atmosphere at room temperature. The base pressure of the system was kept at $1.0 \times 10^{-4}$ Pa. Before deposition, b-Si substrates were cleaned and activated by plasma treatment with the gas mixture of Ar and H$_2$ for 30 min. The deposition pressure was 2.0 Pa. The prepared film was treated in air and under vacuum (~1.0 Pa) at 600 °C for 30 min to form TiO$_2$ protective layer without and with graded oxygen defects. Additionally, varying the deposition time of Ti layer acquired the different concentrations of oxygen defects (its preparation conditions listed in Supplementary Table 1) for evaluating the relationship between PEC behaviors and oxygen defects. To further enhance the efficiency of PEC hydrogen production, Pd NPs were added on the surface of Si-based photocathodes according to the literature[30]. Details of the preparation parameters of each layer are described in Supplementary Table 1.

The back sides of all samples were first polished, and then were coated by Au layer with ~300 nm thickness and bonded with a Cu wire by using silver paint, achieving an ohmic back contact. After drying, the epoxy was used to encapsulate the entire back side and partial front side of the b-Si electrodes, building an exposed active area of ~0.1 cm$^2$. The geometrical area of the exposed electrode surface was determined by calibrated digital images and ImageJ[48].

**Physicochemical characterization**. To explore the crystalline structure of the samples, a Rigaku diffractometer (Rigaku Ultima IV) was employed to perform $2\theta$ XRD scans with the grazing angle of 1° at the scan rate of 2° min$^{-1}$ in Cu Kα radiation ($\lambda = 0.15406$ nm). EPR spectra (JES FA−200) with continuous-wave spectrometer were taken on a sweeping magnetic field and an X band (9.2 GHz) at room temperature. HRTEM (Tecnai G2 F20 S−Twin) were employed to observe the cross section of the film. STEM (Tecnai G2 F20 S−Twin) with a high angle annular dark field (HAADF) detector and EDS were applied to synchronously analyze the component content and structure. The chemical composition was investigated by XPS (Thermo escalab 250XI) with monochromated Al Kα radiation at a pass energy of 29.4 eV. C1s peak (284.8 eV) arising from adventitious carbon was used to calibrate the binding energies. To observe the compositions of each layer of Pd/b2-TiO$_2$/b-Si, XPS depth profiling was conducted via Ar ion beam with a power of 150 W and a beam spot of 500 μm. In this condition, the etching time of ~20 s, ~40 s, and ~75 s corresponded to the etching depth of 5 nm, 10 nm, and 20 nm for Pd/b2-TiO$_2$/b-Si. Fluorescence spectrophotometer (Fls−980, Edinburgh) was used to obtain the room-temperature PL spectra with an excitation wavelength of 405 nm. Time-resolved PL decay spectra were also tested using the same equipment. The PL lifetime of the sample was fitted by using the experimental decay transient data with bi-exponential decay model. The atomic force microscopy (AFM) images presenting both the surface morphology and microscopic C−V curves were collected via Nanocute SII scanning probe microscopy with contact and electric models. The concentration of Ti and Pd in the electrolyte was detected by ICP-MS with Thermo Scientific iCAP-Q instrument. The sample was taken out of the electrolyte using a pipette and the total volume of the electrolyte (90 ml) was measured after each experiment to calculate total mass loss. The sample was diluted to achieve a 0.1 M NaOH solution. The calibration tests were performed using diluted solutions of Ti and Pd (standards with 1000 μg metal/ml). Conversion of the measured amounts of Ti in the electrolyte into monolayers of TiO$_2$ lost during PEC testing was achieved using the lattice distance between two layers assuming a distance of 4.756 Å and a TiO$_2$ density of 3.8 g cm$^{-3}$.

The optical diffuse reflection was monitored on a UV-visible-near-IR spectrophotometer (Hitachi, UV-4100) by an integrating sphere with the normal incidence from 350 to 2600 nm. The Kubelka–Munk theory is generally used for the analysis of diffuse reflection ($R$) spectra to obtain the absorption coefficient ($a$) of the samples as following

$$F(R) = \frac{(1-R)^2}{2R} \cong \alpha \tag{3}$$

where F($R$) is Kubelka–Munk function. Optical energy bandgap of the sample was calculated by using the classical relation of optical absorption

$$\alpha h\nu = B(h\nu - E_g)^m \tag{4}$$

where $E_g$, $B$ and $h\nu$ denote the optical bandgap, band tailing parameter, and photon energy, respectively. The value of $m$ is 2, which means the indirect allowed transition.

**Wettability testing**. The wettability of the samples was estimated by observing the contact angle (CA) of a water droplet (10 μL) on the film surface. The sessile droplets were processed by a contact angle system (Chengde Dingsheng Testing Machine Co. Ltd, JY-82A). The droplet images were captured via a CCD camera with space resolution 1280 × 1024 and color resolution 256 gray levels. The CA value of each sample before and after UV-visible light illumination (produced by a Xe lamp with light current of 16 A) for 30 min was averaged from five measurements.

**Temperature dependent tunneling experiments**. Temperature dependent tunneling experiments were performed on Pd/c-TiO$_2$/b-Si, Pd/b1-TiO$_2$/b-Si and Pd/b2-TiO$_2$/b-Si. Before measuring, each sample was cut into 5 × 5 mm$^2$ and then glued onto one quartz glass by conductive silver paint (Leitsilber 200). Part of dried silver paint on the glass, which is exposed to air, is functioned as one electrode, while another electrode is produced by dotted the silver paint with a diameter of ~100 μm on surface of each sample. Tungsten probes were used to contact a front and back collectors for dry measurements. The substrate bias was scanned from 0 to 1.0 V while monitoring the current density.

**PEC measurement**. A three-electrode cell, including the Si-based working electrode, the counter electrode of Pt wire and the Ag/AgCl reference electrode, was performed for PEC measurements in a PEC 1000 system (PerfectLight Co. Ltd.) with solar light (AM 1.5 G, 100 mW cm$^{-2}$) from a solar simulator (optical fiber source, FX300). The solar simulator intensity was calibrated with a reference silicon solar cell and a readout meter (PerfectLight Co. Ltd., PL-MW 200) before each measurement. 1.0 M aqueous NaOH was set as the electrolyte unless noted otherwise. Liner-sweep voltammetry ($J$–$V$) data were collected using a CHI 630E electrochemical workstation with or without illuminating. The voltage was linearly swept at a scan rate of 0.01 V s$^{-1}$ during $J$–$V$ measurement. Readings for Ag/AgCl

were converted to RHE as below.

$$E(\text{RHE}) = E(\text{Ag/AgCl}) + 0.197 + 0.059 \times \text{pH} \tag{5}$$

The durability experiments ($J$–$t$) of Pd/b2-TiO$_2$/b-Si photocathodes were performed at constant external bias of 0.011 and 0.125 V vs RHE corresponding to the electrolyte of 1.0 M NaOH and 0.5 M H$_2$SO$_4$, respectively.

IPCE measurements were conducted in 1.0 M NaOH aqueous solution at −0.2 V vs RHE. The solar simulator (AM 1.5 G, 100 mW cm$^{-2}$) seamlessly connected a monochromator with optical fiber provided monochromatic illumination at a small light spot. Each testing at each wavelength was the dark of 4 s and then the illumination of 4 s. The current was collected at 10 points per second, and the total time is 120 s. The final 10 points of each light and dark cycle were averaged. The photocurrent was equal to the light current subtracting the dark current. The average value of two experiments determined each plot. IPCE was calculated by the equation

$$\text{IPCE} = \left[ \frac{1240 \times (J_{\text{light}} - J_{\text{dark}})}{\lambda P_\lambda} \right] \times 100\% \tag{6}$$

where $J_{\text{light}}$, $J_{\text{dark}}$, $\lambda$ and $P_\lambda$ denote light current, dark current, certain wavelength, density of light intensity at certain wavelength, respectively.

## Data availability
The data supporting this study are available from the authors on reasonable request.

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

## Acknowledgements

The authors are grateful to the National Natural Science Foundation of China (51402100 and 21573066), the Provincial Natural Science Foundation of Hunan (2016JJ1006 and 2016TP1009), the Shanghai Sailing Program (17YF1429800) and Australian Research Council (DP180100568).

## Author contributions

J.Z., H.Z. and S.W. conceived the ideas, designed the research and oversaw the entire project. J.Z. synthesized the catalysts, conducted photoelectrochemical measurements and analysed the data. Y.L., R.W. and C.X. carried out the preparation of black Si and Raman and EPR characterization. J.Z., H.Z., and S.W. co-wrote the paper and commented on the manuscript. S.P.J. commented and improved the language of the paper. Data availabilityThe data supporting this study are available from the authors on reasonable request.

## Additional information

**Competing interests:** The authors declare no competing interests.

