## [Peer Review File · Nature Communications]

Reviewers' comments:

Reviewer #1 (Remarks to the Author):

The main achievement of the manuscript is a promising stability/durability of the protection layer, however, the characterization, analysis and evaluation is to some extent made more difficult and obscured by the choice of black silicon (b-Si) as photo-absorber due to the surface roughness. The case for the protection layer could probably have been more clearly made with a planar silicon photo-absorber, where corrosion or stability could be convincingly demonstrated for a Pd/b²-TiO₂/Si sample. Then b-Si samples could have been added if they contributed to the case. In its present form the manuscript is quite confusing and not very clearly written, and the PEC performance is disappointing, even though you write: " In short, the crystalline TiO₂ layer with graded oxygen defects supports the Si-based photocathode to achieve an outstanding PEC performance and a high stability synchronously." Which is hardly correct; the PEC performance is certainly not outstanding since hardly any energy is harvested, while the stability is very promising.

I cannot support publication in the prestigious Nature Communications.

A minor detail regarding the reply to my Question #1 about the silicon resistivity: Even though you do not supply details about how the four-point measurement was done, I suspect that the calculated resistivity is off by a factor of approximately 5 (i.e., exactly $n/\ln 2$) off; please consult literature on four-point measurements.

Reviewer #3 (Remarks to the Author):

Reviewers Comment on the revised version submitted to Nat comm

Crystalline TiO₂ protective layer with graded oxygen defects for efficient and stable Si-based photocathode

The manuscript titled "Crystalline TiO₂ Si-based photocathode" deals with the black-Si photocathode for PEC water-splitting and black-TiO₂ surface protection. The resultant photocathode shows excellent stability under harsh pH conditions. The authors have addressed two major revisions with more characterizations and measurements. Overall the manuscript has improved a lot and other than some inherent characteristics of p-Si without p-n junction. Overall, I would recommend the manuscript for publication in prestigious Nature communication with below modifications.

1. As the MS deals only with photocathode the half-cell efficiencies (Solar to hydrogen conversion efficiencies (SHCE)) should be shown. (Not as in Figure S22). For more details pl. refer recent article in Nature Energy, 3, 185–192 (2018), Nano Lett. 15, 2817–2824 and other papers cited therein.
2. It would be better to show the electrochemical characteristics of Pd NPs alone, to obtain the Voc of the cell.
3. Still, the presentation and language can be improved for better readability.

Comments from Referee#1:

Firstly, we are of great gratitude to your kindness for providing us with so much profound knowledge and significant information in all the review processes. Your comments offer us many brand-new viewpoints including the resistivity calculated by four-point measurement, the relationship between the resistivity of Si and doping density, and ICP analysis of the electrolyte post-electrolysis.

Q1) The main achievement of the manuscript is a promising stability/durability of the protection layer, however, the characterization, analysis and evaluation is to some extent made more difficult and obscured by the choice of black silicon (b-Si) as photo-absorber due the surface roughness. The case for the protection layer could probably have been more clearly made with a planar silicon photo-absorber, where corrosion or stability could be convincingly demonstrated for a Pd/b2-TiO₂/Si sample. Then b-Si samples could have been added if they contributed to the case.

Our revision and explanation:

Thank you very much for your instructive suggestion.

Based on your suggestion, the characterization and PEC performance of Pd nanoparticles/black TiO₂/planar Si (Pd/b2-TiO₂/planar Si) have been analyzed and measured to further confirm the characterization and effect of the protection layer. As shown in the revised manuscript, the black TiO₂ layer showed graded oxygen defects. Even though Pd/b2-TiO₂/planar Si photocathode had a lower limiting current than Pd/b2-TiO₂/b-Si photocathode, the black TiO₂ layer still provided a promising stability in 1.0 M NaOH. Thus, the Si with or without nanostructure hardly affects the characterization, analysis and evaluation of the protection layer.

{Characterization and photoelectrochemical profile of cocatalyst/protective layer/b-Si: “..... In addition, the characterization, analysis and PEC measurements of Pd/b-TiO₂/planar Si (substituting planar Si wafer to black Si wafer) further demonstrate the effect of b-TiO₂ protection layer roundly (Supplementary Figs. 26-28).”

Supplementary Information:

Figure 26. (a) XRD pattern of b2-TiO₂/planar Si. (b) The measured total hemispherical optical reflectance of b2-TiO₂/planar Si. (c) Ti 2p spectrum for the surface of Pd/b2-TiO₂/planar Si. (d) Ti 2p spectrum for the inner of Pd/b2-TiO₂/planar Si after the etching depth of 20 nm.

Figure 27. (a) Cross-sectional FESEM image of Pd/b2-TiO₂/planar Si. (b) The backscattered electron image corresponding to (a). (c) AFM topography image (left) and typical *I-V* curves of marked position by the number (right) for Pd/b2-TiO₂/planar

Si. Position 1, 2, 3, 4 and 5 in AFM image corresponding to purple, green, Cambridge blue, orange and yellow I - V curve, respectively.

Figure 28. PEC performance of Pd/b2-TiO₂/planar Si. (a) J - V curves of the sample (scan rate is 0.01 V·s⁻¹) in 1.0 M NaOH under 1 sun illumination. The current density measured in dark is almost horizontal line, namely 0 mA·cm⁻². (b) J - t plot of the sample held at -0.077 V vs RHE in 1.0 M NaOH under 1 sun illumination. The inset is the variations of photocurrent density at -0.077 V vs RHE by chopping light illumination. }

Q2) In its present from the manuscript is quite confusing and not very clearly written, and the PEC performance is disappointing, even though you write: "In short, the crystalline TiO₂ layer with graded oxygen defects supports the Si-based photocathode to achieve an outstanding PEC performance and a high stability synchronously." Which is hardly correct; the PEC performance is certainly not outstanding since hardly any energy is harvested, while the stability is very promising. I cannot support publication in the prestigious Nature Communications.

Our revision and explanation:

Firstly, we are sorry for the poor expressions. We have revised the WHOLE manuscript carefully and tried to avoid any grammar errors or confusing expressions and asked several native English writers to check the paper. We believe that the language is now acceptable for publication. The changes have been highlighted in the updated manuscript.

Even though the photovoltage generated by p-Si is definitely lower than that obtained from n⁺p-Si, p-Si is often used as the photocathode materials due to low cost and good thermal stability. In addition, high quality black n⁺p-Si can be difficult to be formed by metal-catalyzed electroless etching method. Therefore, p-Si that natively had a low photovoltage for HER was chosen as the photocathode in this work. Furthermore, the aim of the manuscript is focusing on decoupling the trade-offs between stability and efficiency of p-Si photocathode by the crystalline TiO₂ with graded oxygen defects. To demonstrate the PEC performance of Pd/b2-TiO₂/b-Si photocathode concisely, Table R1 summarizes the reported PEC performance of Si-based photocathodes without buried p-n junction in recent years. On the basis of the data from Table R1, Pd/b2-TiO₂/b-Si photocathode shows a comparable PEC performance to those p-Si-based photocathodes regardless of the durability and the electrolyte. However,

the inaccurate sentence “..... the crystalline TiO₂ layer with graded oxygen defects supports the Si-based photocathode to achieve an outstanding PEC performance” has been corrected as below. As mentioned in the manuscript, the use of crystalline TiO₂ layer with oxygen defects can effectively decouple the trade-offs between stability and efficiency of p-Si photocathode. The proposed method in this manuscript can make an important contribution to promoting the solar-to-hydrogen conversion and relieving the global warming, which is fit for the scope of Nature Communications.

{Characterization and photoelectrochemical profile of cocatalyst/protective layer/b-Si: “..... the crystalline TiO₂ layer with graded oxygen defects supports the p-Si photocathode to achieve a comparable PEC performance”

Table R1. Comparison of selected representative state-of-the-art p-Si-based photocathodes for HER. V_{OS} is the potential measured at a water reduction current density of 1 mA·cm⁻²; J_{H^+/H_2} is the current density at 0 V vs RHE; J_{lim} is the limiting current of the photocathode under illumination.

Configutation	V_{OS} vs RHE (V)	J_{H^+/H_2} (mA·cm ⁻²)	J_{lim} (mA·cm ⁻²)	Ref.
Pd/b-TiO₂/Si	0.2	8.3	35.3	this work
Pt/Ti/SiO ₂ /Si	0.2	4.5	9	Ji et al., Nat. Mater. 2017, 16, 127
Pt/Ti/SiO ₂ /Si	0.4	5	19	Esposito et al., Nat. Mater. 2013, 12, 562
Mo ₃ S ₄ /Si	0.12	8.8	15	Hou et al., Nat. Mater. 2011, 10, 434
1T-MoS ₂ /Si	0.25	17.6	26.7	Ding et al., J. Am. Chem. Soc. 2014, 136, 8504
CoSe ₂ /Si	0.18	9	–	Basu et al., Angew. Chem. Int. Ed. 2015, 54, 6211
Pt/Ti/SrTiO ₃ /Si	0.4	15	35	Ji et al., Nat. Nanotechnol. 2014, 10, 84
b-Si	-0.18	–	36	Oh et al., Energy Environ. Sci. 2011, 4, 1690
NiFe LDH/Ti/Si	0.3	7	28	Zhao et al., ACS Energy Lett. 2017, 2, 1939

Q3) A minor detail regarding the reply to my question #1 about the silicon resistivity: Even though you do not supply details about how the four-point measurement was done, I suspect that the calculated resistivity is off by a factor of approximately 5 (i.e., exactly $\pi/\ln 2$) off; please consult literature on four-point measurements.

Our revision and explanation:

Thanks a lot for pointing out our mistake and providing a useful advice.

Firstly, we used a wrong equation to calculate the silicon (Si) resistivity in the previous response. According to your recommendation, we carefully consult the literatures on four-point measurements and recalculate the Si resistivity as follows.

Methods: “.....p-type Si wafer with a resistivity of 2–4 $\Omega \cdot \text{cm}$ (Supplementary Figs. 33 and 34) was first cleaned sequentially

Supplementary Information:

Figure 33. (a) Schematic of a four-point measurement with line-up tips forming the contacts on the surface of the planar Si. (b) Equidistant four-probe current-voltage curves of the planar Si.

As shown in Supplementary Fig. 33a, the thickness of Si wafer (d), the tip-tip distance (S) and the distance between tip and sample edge (L) are 0.5, 1 and 2 mm, respectively. The resistivity (ρ) of Si wafer is calculated as follows:

$$\rho = \frac{2\pi S}{B_0} \cdot \frac{V}{I}$$

where I is the applied current, V the corresponding voltage, S the tip-tip distance and B_0 the correction factor (3.104) determined by S/d and L/S . Based on the resistance (R) ranged from 10 to 20 Ω , the real resistivity of Si is around 2-4 $\Omega \cdot \text{cm}$.

In addition, the Mott-Schottky plot of the planar Si has been added in the updated manuscript.

Supplementary Information:

Figure 34. Mott-Schottky plot of the planar Si from capacitance measurement as a function of potential vs RHE under dark conditions.

The Mott-Schottky plot was acquired at a frequency of 1 KHz in 0.5 M H_2SO_4 solution by a CHI 660 potentiostat. The Mott-Schottky equation is shown below:

$$\frac{1}{C^2} = \frac{2}{q\epsilon_s\epsilon_0 A^2 N_D} (V - V_{fb} - kT/q)$$

where C is capacitance, q the charge of an electron (1.60×10^{-19} C), ϵ_0 the vacuum permittivity (8.85×10^{-14} F·cm⁻¹), ϵ_s the permittivity of silicon (1.05×10^{-14} F·cm⁻¹), A the area of the sample, N_D the donor density, V the applied bias, V_{fb} the flat band voltage, k Boltzmann's constant (1.38×10^{-23} J·K⁻¹), and T the temperature (25 °C). The x-intercept of the Mott-Schottky plot was reached at the bias that needs to be applied to cause the bands to become flat. Also, the slope of the plot can be used to calculate the donor density of the electrode. The x-intercept plus kT/q (~0.025 V) equals the flat band voltage. The N_D can be calculated using the equation:

$$N_D = \frac{2}{\epsilon\epsilon_s\epsilon_0} \left[\frac{d(1/C^2)}{dV} \right]^{-1}$$

where ϵ and $\left[\frac{d(1/C^2)}{dV} \right]$ are the dielectric constant of silicon (11.68) and the slope of

the sharp increase from 0-0.12 V region. Thus, N_D for the planar Si can be calculated to be 3.21×10^{15} cm⁻³, corresponding to the resistivity of Si wafer (2-4 Ω·cm) basically.

Comments from Referee#3:

The manuscript titled "Crystalline TiO₂ Si-based photocathode" deals with the black-Si photocathode for PEC water-splitting and black-TiO₂ surface protection. The resultant photocathode shows excellent stability under harsh pH conditions. The authors have addressed two major revisions with more characterizations and measurements. Overall the manuscript has improved a lot and other than some inherent characteristics of p-Si without p-n junction. Overall, I would recommend the manuscript for publication in prestigious Nature Communications with below modifications.

We are of great gratitude to your positive appraisal and instructive comments. We have revised our manuscript and provided a point-by-point response to your comments as follows:

Q1) As the MS deals only with photocathode the half-cell efficiencies (Solar to hydrogen conversion efficiencies (SHCE)) should be shown. (Not as in Figure S22). For more details pl. refer recent article in Nature Energy, 3, 185-192 (2018), Nano Lett. 15, 2817-2824 and other papers cited therein.

Our revision and explanation:

Thank you very much for the articles.

According to the methods reported by the articles, the solar-to-hydrogen conversion efficiencies (SHCE) has been provided in the updated manuscript as below:

{**Characterization and photoelectrochemical profile of cocatalyst/protective layer/b-Si**: “..... But regrettably, the solar-to-hydrogen (STH) efficiency and solar-to-hydrogen conversion efficiency (SHCE) of Pd/b2-TiO₂/b-Si (Supplementary Figs. 21 and 22) were low because of the lack of p-n junctions.”

Supplementary Information:

Figure 22. Electrochemical characterization of Pd nanoparticles deposited on ITO in a 1.0 M NaOH electrolyte in dark. The inset is the characteristics of Pd/b2-TiO₂/b-Si photocathode in a 1.0 M NaOH electrolyte under simulated AM 1.5 G illumination. V_{OC} is the open-circuit voltage of the photocathode; V_{OS} is the potential measured at a water reduction current density of $1 \text{ mA}\cdot\text{cm}^{-2}$; E^0 is the equilibrium water reduction potential in the 1.0 M NaOH electrolyte, which is 0 V vs RHE; J_{H^+/H_2}^0 is the current density at E^0 ; FF is the fill factor of the photocathode; and $SHCE$ is the solar-to-hydrogen conversion efficiency of the photocathode.

The preparation conditions of Pd nanoparticles on ITO are identical to those in b2-TiO₂/b-Si. The V_{OC} is calculated as follows:

$$V_{OC} = V_{ph,10} - V_{ca,10}$$

where $V_{ph,10}$ is the potential of Pd/b2-TiO₂/b-Si photocathode at a current density of $10 \text{ mA}\cdot\text{cm}^{-2}$ under illuminated, and $V_{ca,10}$ the potential of Pd nanoparticles on ITO at a current density of $10 \text{ mA}\cdot\text{cm}^{-2}$ in dark. The $SHCE$ is calculated as follows:

$$SHCE = \frac{|V_{OS} - E^0| J_{H^+/H_2}^0 FF}{P_{incident}}$$

where $P_{incident}$ is the illumination power density.

Q2) It would be better to show the electrochemical characteristics of Pd NPs alone, to obtain the V_{OC} of the cell.

Our revision and explanation:

Thank you for your kind suggestion.

Based on your useful guidelines, the electrochemical characteristics of Pd nanoparticles deposited on ITO are used to obtain the open-circuit voltage (V_{OC}) of

the Pd/b2-TiO₂/b-Si photocathode:

{Supplementary Information:

Figure 22. Electrochemical characterization of Pd nanoparticles deposited on ITO in a 1.0 M NaOH electrolyte in dark. The inset is the characteristics of Pd/b2-TiO₂/b-Si photocathode in a 1.0 M NaOH electrolyte under simulated AM 1.5 G illumination. V_{OC} is the open-circuit voltage of the photocathode; V_{OS} is the potential measured at a water reduction current density of $1 \text{ mA}\cdot\text{cm}^{-2}$; E^0 is the equilibrium water reduction potential in the 1.0 M NaOH electrolyte, which is 0 V vs RHE; J_{H^+/H_2}^0 is the current density at E^0 ; FF is the fill factor of the photocathode; and $SHCE$ is the solar-to-hydrogen conversion efficiency of the photocathode.

The preparation conditions of Pd nanoparticles on ITO are identical to those in b2-TiO₂/b-Si. The V_{OC} is calculated as follows:

$$V_{OC} = V_{ph,10} - V_{ca,10}$$

where $V_{ph,10}$ is the potential of Pd/b2-TiO₂/b-Si photocathode at a current density of $10 \text{ mA}\cdot\text{cm}^{-2}$ under illuminated, and $V_{ca,10}$ the potential of Pd nanoparticles on ITO at a current density of $10 \text{ mA}\cdot\text{cm}^{-2}$ in dark. The $SHCE$ is calculated as follows:

$$SHCE = \frac{|V_{OS} - E^0| J_{H^+/H_2} FF}{P_{incident}}$$

where $P_{incident}$ is the illumination power density.

Q3) Still, the presentation and language can be improved for better readability.

Our revision and explanation:

We are sorry for our poor expressions.

We have revised the WHOLE manuscript carefully and tried to avoid any grammar errors or confusing expressions and asked several native English writers to check the paper. We believe that the language is now acceptable for publication. The changes have been highlighted in the updated manuscript.

REVIEWERS' COMMENTS:

Reviewer #1 (Remarks to the Author):

Even though there still are remaining language issues, the manuscript has improved such that it can now be published.

Reviewer #3 (Remarks to the Author):

Crystalline TiO₂ protective layer with graded oxygen defects for efficient and stable Si-based photocathode

The reviewer is fully satisfied with the reply by the authors and after multiple revisions and improvements, now the manuscript looks fit to be published. The reviewer recommends the manuscript for publication in the Nature communications.

Response to the Comments

Reviewer #1 (Remarks to the Author):

Even though there still are remaining language issues, the manuscript has improved such that it can now be published.

Our revision and explanation:

Firstly, we are of great gratitude to your help for improving our work in the whole review processes. We are very pleased for receiving a few positive comments from you.

Additionally, based on your comments, we ask a colleague, Prof. Sanping Jiang, who speaks English in almost 30 years and works in Curtin University, to correct the language issues in our manuscript. The changes of the language are tracked by the 'track changes' feature.

Reviewer #3 (Remarks to the Author):

Crystalline TiO₂ protective layer with graded oxygen defects for efficient and stable Si-based photocathode

The reviewer is fully satisfied with the reply by the authors and after multiple revisions and improvements, now the manuscript looks fit to be published. The reviewer recommends the manuscript for publication in the Nature Communications.

Our revision and explanation:

Thank you very much for your instructive comments in the whole review processes.